# Effect of Seawater Irrigation on the Sugars, Organic Acids, and Volatiles in 'Reliance' Grape

**Menglong Liu [1], Meng Yu [1], Yuxin Yao [1], Heng Zhai [1], Meiling Tang [2], Zhen Gao [1,\*] and Yuanpeng Du [1,\*]**

[1]  State Key Laboratory of Crop Biology, Collaborative Innovation Center of Fruit and Vegetable Quality and Efficient Production, College of Horticulture Science and Engineering, Shandong Agricultural University, Tai'an 271018, China; mlliu@ibcas.ac.cn (M.L.); simpletake00@163.com (M.Y.); yaoyx@sdau.edu.cn (Y.Y.); zhaih@sdau.edu.cn (H.Z.)

[2]  Department of viticulture, Yantai Agricultural Science Academy, Yantai 264003, China; tmling1999@163.com

\*  Correspondence: gaoz89@sdau.edu.cn (Z.G.); duyp@sdau.edu.cn (Y.D.)

**Abstract:** Ongoing climate change in recent decades exacerbated the decline in agricultural water use, and seawater irrigation could feasibly alleviate the shortage of water resources, which restricts viticulture in some countries. However, studies on the effects of seawater irrigation on grape volatiles are limited. Herein, 'Reliance' grapevines were irrigated with diluted seawater (10% concentrations) in the field since the pea-size berry stage (S1), stage EL 32 (S2), and the pre-veraison period (S3) every seven days. Results showed irrigation with seawater significantly increased the sugar content and decreased the organic acids when compared with the control berries. Seawater irrigation did not induce secondary soil salinization, and it enhanced the volatiles in the fatty acid and isoprene pathways without affecting the amino acid pathway aroma. More terpenes were found in seawater-treated berries, including citronellol, β-myrcene, α-terpineol, and trans-rose oxide. Gene profiling by RT-qPCR analysis revealed that *VvLOXA* could be the primary gene in C6 volatile biosynthesis altered by the seawater. Moreover, seawater irrigation during the pea-size period had the best effect on fruit quality. This work adds to our understanding of the effect of seawater irrigation on grape aroma quality and provides a sufficient basis for the utilization of seawater in vineyards.

**Keywords:** irrigation; volatile composition; soil

## 1. Introduction

Grapes are a popular fruit consumed worldwide and products from grape processing, such as raisins and wine, also have high economic value. Although grapevines have substantial drought tolerance [1], the shortage of water resources due to global warming, increasing water tax, pollution, and other problems restrict the full exploitation of the grapevine. In this case, the utilization of diluted seawater for vineyard irrigation is a water-saving strategy worth exploring. Seawater is approximately 3.5% saline, and the significant elements are sodium, magnesium, calcium, and potassium, inducing salinity stress in plants and negative effects on the soil [2]. Research on the effects of diluted seawater treatments and salt stress on plants has made significant progress in various crops, including watermelon, apple [3], and grape [4]. Previous studies demonstrated that irrigation with diluted seawater resulted in the accumulation of soluble sugars, polyphenols, carotenoids, and fatty acids in leafy vegetable crops [5]. For instance, tomato plants irrigated with diluted seawater produced berries characterized by high amounts of vitamin C, vitamin E, and dihydrolipoic acid [6]. Another study evaluated the effects of long-term NaCl-treatment on grapes and found that moderate salinity (20 and 60 mM) enhanced the overall berry quality but decreased the aroma. However, high salinity (100 and 150 mM) decreased berry quality [7]. Our previous work demonstrated that field irrigation with seawater improves fruit quality, and more than four consecutive years of seawater irrigation did

not induce secondary soil salinization. These findings prove that seawater irrigation could provide a feasible and beneficial irrigation strategy in viticulture [2].

At present, the effect of diluted seawater on grape volatiles is obscure. As one of the well-known fruits globally, grapes have hundreds of volatiles [8,9], and their concentrations, properties, and balances between each other determine the features of different grape cultivars and wines [10,11]. Grape volatiles can be divided into terpenoids, norisoprenoids, aromatic compounds, aliphatic volatile compounds, methoxypyrazines, and organo-sulfur compounds [12]. These biomolecules can be further classified into three categories according to the types of precursors and the synthetic routes: isoprene pathway, fatty acid pathway, and amino acid pathway volatiles. Straight-chain fatty acid derivatives are also called green leaf volatiles (GLVs); they include C6 aldehydes, C6 alcohols, and C6 esters, all of which are generated through the lipoxygenase pathway from $C_{18}$-polyunsaturated fatty acids (α-linolenic acid and linoleic acid). Various genes, including *VvLOX* (lipoxygenase), *VvHPL* (hydroperoxides lyase), *VvADH* (alcohol dehydrogenase), and *VvAAT* (alcohol acyltransferases), are key in this fatty acid metabolism [12–14]. Generally, the fruity and floral scent of these GLVs, especially C6 esters, is the primary source of the fruity and floral scent of aroma in both grapes and wine [15,16]. Moreover, these volatiles have also been shown to serve as essential signal molecules for the pollination and/or prevention of pathogen intrusion [17–19]. Furthermore, as a product in the isoprene pathway, terpenes are considered primary compounds of Muscat cultivars, such as Shine Muscat and Thermal Spring [20]. Amino acid pathway volatiles also contribute to the aromatic characteristics of grapes; benzene acetaldehyde has a hyacinth-like smell, and methoxy pyrazines (MPs), found in immature Cabernet Sauvignon grapes, has a vegetal aroma.

Therefore, this study aimed to determine the effect of 10% seawater irrigation starting at different grapevine phenological stages on grape volatiles. The 'Reliance' grape (Ontario × Suffolk Red), which is abundant in fruity but little floral scent, was selected, and its volatiles were analyzed according to the three synthetic routes. This experiment will illuminate on the changes in grape volatiles following diluted seawater treatment and provide feasible suggestions on the potential of applying seawater irrigation to enhance viticulture.

## 2. Materials and Methods

### 2.1. Plant Materials and Treatments

Five-year-old 'Reliance' grapes (*V. labrusca* × *V. vinifera.*, Ontario × Suffolk Red) grown in a vineyard in Tai'an, Shandong province, China (average annual rainfall was 638.4 mm between 2014 and 2019, and meteorological data for the vineyard in 2018 is available in Table S1), were irrigated with seawater in 2018. The vines were planted in a vertical trellis system and spaced at 2.5 m × 1 m in a north-south row orientation. A furrow was made 40 cm away from the main trunk on both sides to ease treating diluted seawater. All treatments were replicated thrice in a randomized block design. Three blocks with similar texture and fertility in each experimental block were selected and randomly assigned S1, S2, and S3. A control group was also included. Fifty vines selected in every two rows formed a replicate. Guard rows were set between treatments to reduce the marginal effect.

Irrigation treatment was followed according to our published method with little modification [2]. Briefly, the vines were irrigated with 10% seawater every seven days starting at the pea-size period (S1), stage EL 32 (S2), and pre-veraison period (S3), seawater was irrigated six times for S1, five times for S2, and four times for S3. Seawater was drawn from Penglai city, China. The physical and chemical properties of the undiluted seawater are outlined in Table 1. The initial irrigation time is presented in Table 2. The control group was irrigated with fresh water (EC = 50 us/cm). Each irrigation was based on weather forecasts approximately two days before the rain. The experiment used the method of furrow irrigation; the furrow was 40 cm wide and 100 cm long for each vine, the average irrigation amount per plant was 20 L. Vines were slightly hoed and covered with soil after irrigation to reduce the evaporation of water.

**Table 1.** Physical and chemical properties of original concentration seawater.

| Na$^+$ (mg L$^{-1}$) | Ca$^{2+}$ (mg L$^{-1}$) | Mg$^{2+}$ (mg L$^{-1}$) | Cl$^-$ (mg L$^{-1}$) | SO4$^{2-}$ (mg L$^{-1}$) | Degree of Mineralization (g L$^{-1}$) | pH |
|---|---|---|---|---|---|---|
| 11,041 | 300 | 1627 | 19,241 | 2498 | 1200 | 7.9 |

**Table 2.** Specific date of initial irrigation time and sampling time in 2018.

| Date | May 29th | May 31st | Jun. 12th | Jun. 14th | Jun. 26th | Jun. 28th | Jul. 5th | Jul. 12th | Jul. 19th |
|---|---|---|---|---|---|---|---|---|---|
| initial irrigating time | Control, S1 | | S2 | | S3 | | | | |
| sampling time | | EL 31 | | EL 33 | | EL 35 | EL 36 | EL 37 | EL 38 |

Samples were collected at six growth stages: EL 31 (pea-size berries), EL 33 (still hard and green berries), EL 35 (berries begin to color and enlarge), EL 36 (berries with intermediate Brix values), EL 37 (berries not quite ripe), and EL 38 (harvest-ripe berries) (Coombe, 1995) (Table 2). Samples were used to determine berry maturity parameters, sugars, organic acids, volatiles, and gene expression.

At each stage, 600 berries of each replicate were manually collected in iceboxes and transported to the laboratory within 30 min. Fifty replicates in each group were used to determine the berry hardness within 2 h after harvesting. The berries were immediately frozen in liquid nitrogen and stored at −80 °C after weighing to determine sugars, organic acids, volatiles, and RT-qPCR analysis. Soil samples were collected from the soil horizon (0–40 cm) 40 cm away from grapevines at the EL 38 stage for the determination of soluble salt, chloride (Cl$^-$), and sulfate (SO4$^{2-}$) content, and the pH level. The analyzed samples were a mixture of soils from twelve grapevines from each replicate. All treatments were carried out in triplicate and results presented as the average of three analyses ($n = 3$).

Then, the determination of berry weight, hardness, total soluble sugar, total soluble solids (TSS), and titratable acidity (TA) was conducted. A scale of one percent accuracy was used to measure the average berry weight of 100 berries. Texture parameters of the berries (skin hardness, frangibility, firmness, and average flesh hardness) were determined using a Universal Testing Machine Texture Analyser (TA.XT Plus, Stable Micro Systems, Surrey, UK) equipped with a P/2n (d = 2 mm) needle probe and an HDP/90 platform. The penetration speed was 1 mm s-1, the puncture depth was 7 mm, and the minimum perceptive power was 5 g. Fifty fresh berries were randomly selected from each replicate for these analyses. The equatorial center of the berry was selected as the puncture point [21]. The total soluble sugar, total soluble solids (TSS), and titratable acidity (TA) were determined on the extracted fruit juice as described by Li et al. [7].

*2.2. Soluble Sugar and Organic Acid Compound Determination*

Soluble sugar and organic acid compounds were extracted and determined using a capillary electrophoresis system (Beckman P/ACE, Fullerton, CA, USA) according to our previously published method [7].

*2.3. Extraction of Aromatic Compounds and Analysis*

The extraction and determination of volatile compounds followed the method described by Li [7] with minor modifications. Briefly, 100 g of de-seeded grape berries were ground into powder in liquid nitrogen followed by the addition of 0.5 g of PVPP (Crosslinked Polyvinylpyrrolidone, to inhibit oxidation of phenols) and 0.5 g of D-glucose acid lactone (to inhibit glycosidase activity). The mixture was left to stand for 4 h at 4 °C and then centrifuged at 4000 rpm at 4 °C for 15 min to collect the supernatant (juice). Subsequently, 10 mL of juice was mixed with 1 g of NaCl powder (to prevent sample browning) and 3 μL of the internal standard (2-octanol, 0.822 g L$^{-1}$) in a 20 mL vial capped with a PTFE-silicon septum.

An auto headspace solid-phase microextraction (HS-SPME) was then used to extract volatile compounds on an AOC-6000 autosampler (Shimadzu, Kyoto, Japan). The SPME fiber (50/30 μm CAR/DVB/PDMS, Supelco, Bellafonte, PA, USA) was preconditioned in the injection port of the GC before the extraction following the manufacturer's instructions. The vial contents were continuously stirred for 30 min at 35 °C followed by exposure of the SPME fiber to the headspace for 40 min while maintaining the sample at 35 °C. After sampling, the SPME fiber was then inserted into the GC injector after sampling and left for 5 min at 250 °C to allow for thermal desorption of the analytes.

A GCMS-TQ8050 system (Shimadzu, Kyoto, Japan) was employed to analyze the volatile compounds. A capillary column (RTX-5 MS, 60 m × 0.25 mm × 0.25 μm, Shimadzu, Kyoto, Japan) was used to separate the volatiles at a1 mL min$^{-1}$ flow rate of the carrier gas (Helium). The injection mode was splitless at an injector temperature of 250 °C. The oven temperature was set at 40 °C for 2 min, then heated to 230 °C at a speed of 5 °C min$^{-1}$, and finally maintained at 230 °C for 5 min. The mass spectrometer interface and ion source temperature was set at 250 °C and 230 °C, respectively. The mass spectra were acquired using an electron ionization mode (EI) at an electron multiplier voltage of 70 eV and a full mass scan of $m/z$ 30–450. Each compound was identified using the NIST and Wiley 2 libraries. A series of C7-C27 n-alkane standards (Supelco, Bellefonte, PA, USA) were determined under similar chromatographic conditions to calculate their retention indices (RI). Volatile compounds were identified by comparing their RI and mass spectra with the NIST2011 library. A synthetic juice matrix (pH = 3.2, citrate–phosphate buffer solution) was prepared for the qualification of volatile compounds. Standards were first dissolved in methanol, mixed, and then added into the synthetic juice matrix. The standards were then diluted to 15 levels, extracted and analyzed using the procedure used to analyze the grape samples. Volatile compounds without corresponding standards were semi-quantified using the internal standard (2-octanol, 0.822 g L$^{-1}$). Contents of the volatile compounds were expressed as ng/g of the fresh weight. The calibration curves of the standards are outlined in Table S2.

### 2.4. Determination of Soluble Salt Content, $Na^+$, $Cl^-$, and $SO4^{2-}$ Content, Moisture Content, and pH of Soil

The total soluble salt content in the soil was determined using the residue drying-mass method. The soil leachate (20–50 mL; 1:5) was placed in a 100 mL porcelain evaporation dish with a known dry weight and then boiled in a water bath, dropping $H_2O_2$ (150 g mL$^{-1}$) around the dish to moisture the residual. The treatment was then repeated with $H_2O_2$ until the organic matter was completely oxidized. As such, the dry residue was all white. The residue was then placed in the oven (105–110 °C) after evaporation to further dry for 1–2 h. Once cooled it was weighed with an analytical balance. The evaporating dish and residue was dried again for 0.5 h, cooled in a desiccator and then weighed. Each sample was measured in triplicate. The soil moisture content was determined using the drying method while the $Na^+$ content was determined through atomic absorption spectroscopy (Perkin Elmer AA300, PerkinElmer Inc., Waltham, MA, USA). $Cl^-$ concentration was determined using the silver nitrate titration method while the $SO4^{2-}$ content was determined using the EDTA indirect titrimetric method. The pH of the soil and soil leachate (1:1) was measured in triplicate using a potentiometer and acidometer (Sartorius, PB-10, 37079 Goettingen, Germany), respectively.

### 2.5. Real-Time Quantitative PCR Assays (RT-qPCR)

The random selection of 20 berries was carried out followed by deseeding and grinding of the remaining parts into powder in liquid nitrogen. Total RNA of the ground samples was subsequently extracted using the FastPure Plant Total RNA Isolation Kit (RC401, Vazyme, Nanjing, China). First-strand cDNA was synthesized from 1 μg of the total RNA using the PrimeScriptTM RT reagent Kit with gDNA Eraser (TaKaRa, Dalian, China) following the manufacturer's instructions. Quantitative PCR was carried out on a real-time PCR system

(Bio-Rad iQ5) using the SYBR Green Kit (TaKaRa, Dalian, China). The thermal cycling conditions used were similar to those published by Sun et al. [22]. The primers used are listed in Table S3.

### 2.6. Statistical Analysis

Statistical analysis was performed using the SPSS (V20.0, IBM, Armonk, NY, USA) software. Data normalization and principal component analysis (PCA) were also performed using SPSS. Figures were prepared using Microsoft Excel 2016. Differences were analyzed by a two-way ANOVA followed with a Tukey multiple comparison post-hoc test at $p < 0.05$.

## 3. Results

### 3.1. Effect of Seawater Irrigation on Average Weight and Texture Parameters of the Berry

The average berry weight of the berries showed an increasing trend during the entire growing stage and plateaued after the EL 37 stage. No significant differences were found between the average berry weight of berries in the 10% seawater treatment and the control at different developmental phases (Figure 1).

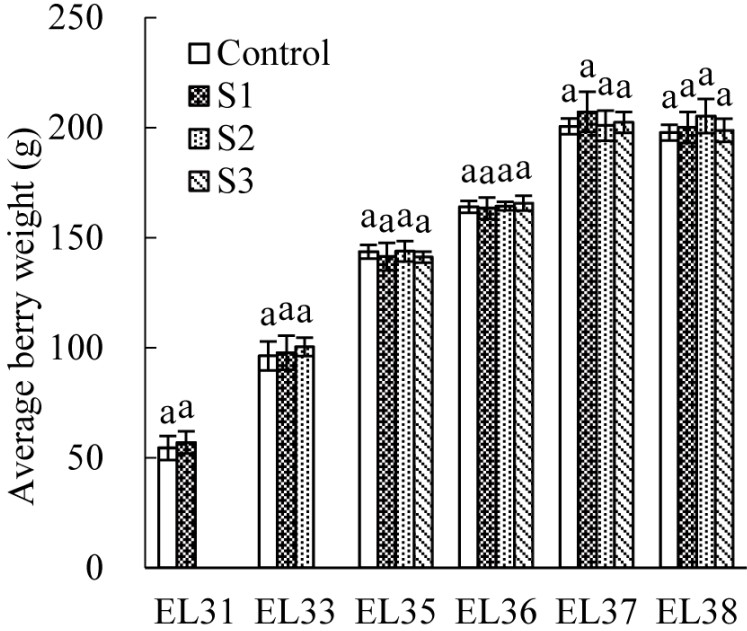

**Figure 1.** Effect of seawater irrigation on the average berry weight of 'Reliance' grapes. Values are the means of three replicates, and error bars denote the SD. Different lowercase letters indicate significant differences of samples in one developmental phase, according to Tukey multiple comparison post-hoc test at $p < 0.05$.

The berries in stages EL 31 and 33 were too small for texture detection; thus, only EL 35 to EL 38 were used to evaluate the skin hardness, frangibility, firmness, and average hardness of the berry flesh (Figure 2). No significant differences in skin hardness were detected between the control and 10% seawater treatments (Figure 2a), while S1 and S2 treatments significantly increased the average hardness of flesh at EL 38 (Figure 2d). This finding could be attributed to the skin of "Reliance" berries being so thick that the slight differences in the skin hardness of different groups were covered. However, the S1 treatment significantly reduced the berry frangibility at EL 35 and EL 36, although the difference disappeared at EL 37 and EL 38 (Figure 2b). This result shows that S1 treatment promotes berry softening earlier than other treatments, thereby reducing the frangibility of the berry in the early stages of fruit development. Seawater treatment improved the berry firmness at EL 38 (Figure 2c). The S1 and S2 treatments had the most significant effect on improving the fruit firmness of "Reliance" berries, which were 26.60% and 23.69% higher

than the control berries. Furthermore, the berry firmness of the S3 treatment was increased by 18.75% compared to the control berries.

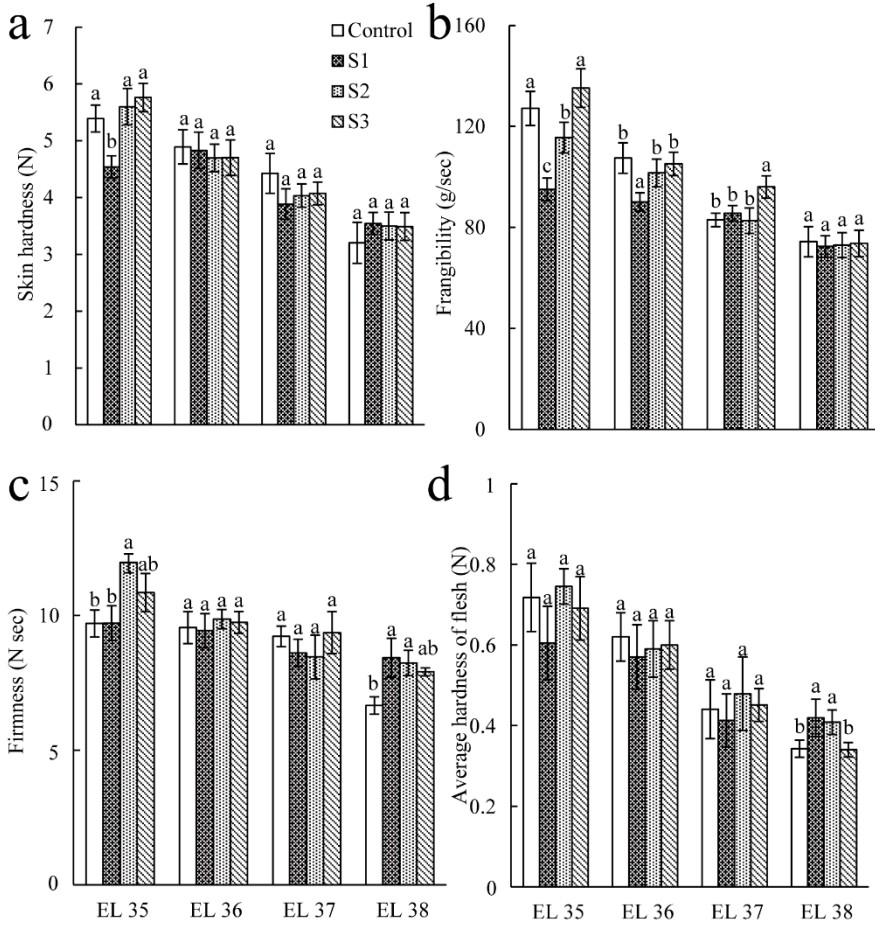

**Figure 2.** Effect of seawater irrigation on the texture of 'Reliance' grapes. (**a**) Skin hardness, (**b**) frangibility, (**c**) firmness, (**d**) average hardness of flesh. Values are the means of fifty replicates, and error bars denote the SD. Different lowercase letters indicate significant differences of samples in one developmental phase, according to Tukey multiple comparison post-hoc test at $p < 0.05$.

### 3.2. Effect of 10% Seawater Irrigation on the Concentration of Sugar and Organic Acids

As shown in Figure 3, the content of total soluble solids (TSS), soluble sugar, glucose, and fructose increased continuously during the growth and maturity of "Reliance" berries. Overall, 10% seawater irrigation in different periods enhanced the content of TSS, soluble sugar, glucose, and fructose in treated berries relative to the control berries. At EL 38, the TSS content of grapes in S1, S2, and S3 treatments increased by 1.87%, 1.80%, and 0.74%, respectively, compared with the control berries (Figure 3a); the soluble sugar content increased by 33.77%, 21.43%, and 15.69%, respectively (Figure 3b); the glucose content increased by 23.62%, 16.69%, and 14.31%, respectively (Figure 3c); the fructose contents increased by 38.69%, 34.65%, and 22.78%, respectively (Figure 3d). More importantly, the S1 treatment significantly affected the TSS, soluble sugar, glucose, and fructose content in the fruits from the EL 33 period; the S2 treatment had a significant impact on the TSS, soluble sugar, glucose, and fructose content in the fruit from the EL 35 period. For the S3 treatment, significant differences first appeared at EL 36. This result indicates that the sooner 10% seawater is irrigated, the earlier the sugar content is improved. In the last period, EL 38, berries of the S1 group showed higher content of TSS, soluble sugar, glucose, and fructose compared with the S2, S3, and the control groups.

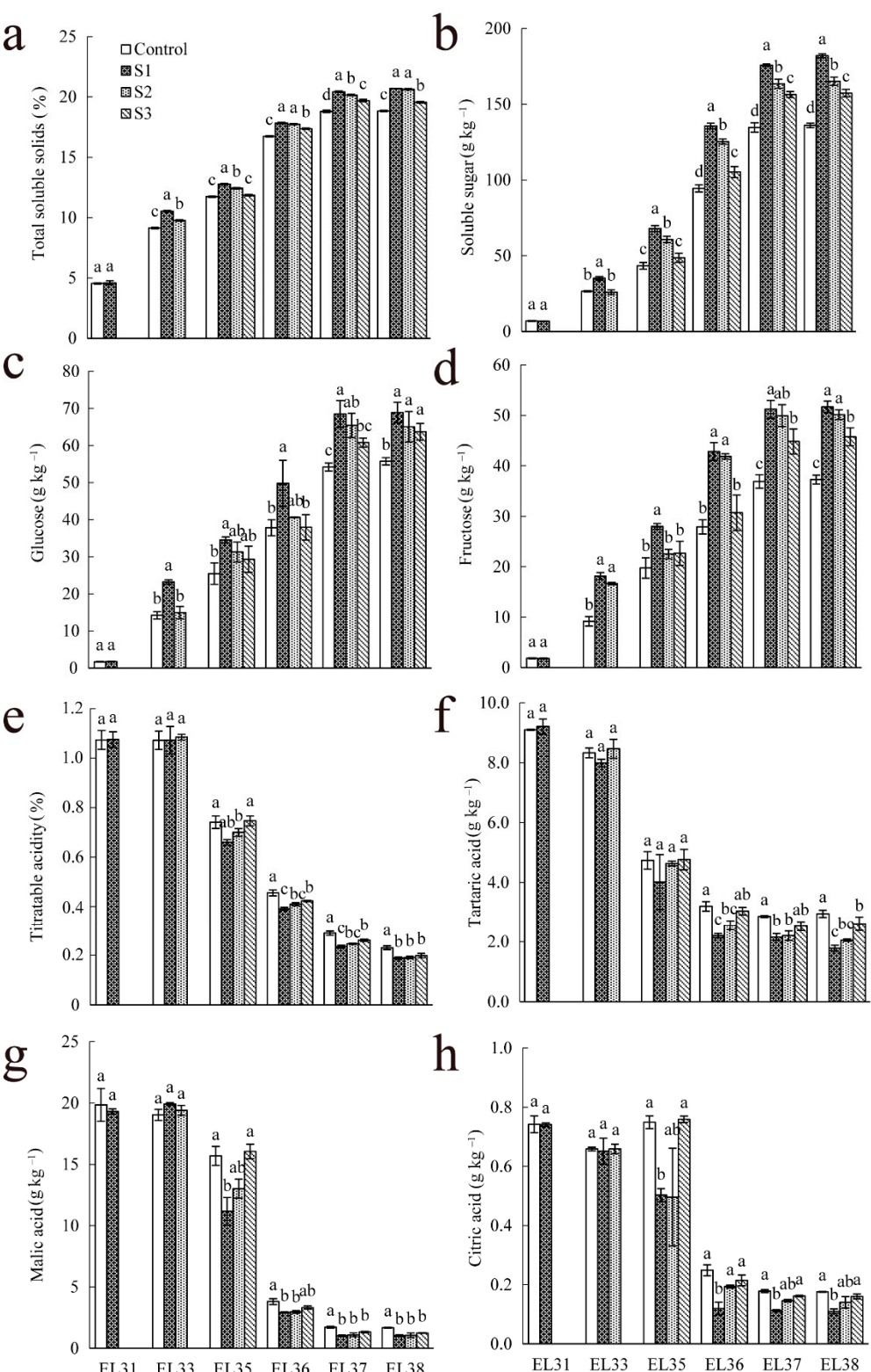

**Figure 3.** Effect of seawater irrigation on the sugar and organic acid content of the 'Reliance' grape. (**a**) Total soluble solids, (**b**) soluble sugar, (**c**) glucose, (**d**) fructose, (**e**) titratable acid, (**f**) tartaric acid, (**g**) malic acid, (**h**) citric acid. Different lowercase letters indicate significant differences of samples in one developmental phase, according to Tukey multiple comparison post-hoc test at $p < 0.05$.

The titratable acidity decreased during the entire growth of "Reliance" berries, with malic acid, tartaric acid, and citric acid showing similar trends (Figure 3e–h). Malic acid was the most abundant organic acid in the early stages (EL 31–35), tartaric acid was the primary organic acid in the later stage of veraison to maturity (EL 36–38), and the content of citric acid was less than 1 g kg$^{-1}$ over the whole course of fruit development. More importantly, 10% seawater irrigation significantly decreased the content of titratable, malic, tartaric, and citric acids in berries at EL 36–38. The significant differences as the consequence of 10% seawater treatment began from EL 35 or EL 36. By EL 38, the content of titratable, malic, tartaric, and citric acids in S1, S2, and S3 groups were significantly lower than the control group. The most significant reduction was in the S1 group, S2 was the second, while S3 had the least. At EL 38, the content of titratable acid in S1, S2, and S3 berries were 18.28%, 16.98%, and 13.59% lower than the control, respectively (Figure 3e); the tartaric acid was decreased by 36.44%, 25.13%, and 11.48%, respectively (Figure 3f); the malic acid levels were reduced by 34.51%, 30.72%, and 25.79%, respectively (Figure 3g); the content of citric acid were 37.61%, 20.13%, and 8.65% lower than the control berries, respectively (Figure 3h).

### 3.3. Effect of Seawater Irrigation on Aroma in 'Reliance' Grape at Growing Stages

The volatiles of "Reliance" berries during the growing stages were analyzed to explore the effect of 10% seawater irrigation on the aroma of grapes. From Figure 4a, it can be concluded that during EL 31~38, the total aroma first decreased slightly and then increased during fruit development from EL 35. In general, the 10% seawater treatment significantly increased the total aroma of berries relative to the control berries. Treatments S1 and S2 significantly increased the fruit aroma content from stage EL 36, but the volatiles of treatment S3 were not significantly different from the control. At 38, the total berry aroma in the S1 and S2 groups was increased by 26.94% and 14.13%, respectively, compared with the control. Moreover, the total aroma of S3 berries was decreased by 4.14% relative to the control.

The aroma of "Reliance" berries was classified into three categories according to the precursors and synthetic routes: isoprene pathway aroma, fatty acid pathway aroma, and amino acid pathway aroma. Evaluation of the aroma concentrations according to the three pathway categories revealed an initial rise and then a decrease, with a similar trend for the total aroma (Figure 4b–d). Of note is the decrease in the isoprene and amino acid pathways aromas that occurred at EL 36 and 37, but which occurred at EL 33 and 35 for the fatty acid pathway aroma. At EL 38, treatments S1 and S2 significantly increased the aroma content of isoprene and fatty acid pathways, relative to the control. No obvious trend was found in the aroma of the amino acid pathway between treatment S1, S2, S3, and the control.

The concentrations of C6 aldehydes, C6 esters, hexanal, and ethyl acetate were measured to further analyze the dynamic changes of representative volatiles of "Reliance" berries (Figure 4e–h). As the main volatiles synthesized in the upstream of the fatty acid pathway, the trends of C6 aldehydes were similar to the fatty acid pathway aroma (Figure 4e); C6 esters, showed a continuously rising trend over the entire course of fruit development (Figure 2f). At EL 38, the concentrations of C6 aldehydes in S1, S2, and S3 groups were significantly increased by 12.50%, 13.71%, and 4.06% relative to the control group, respectively. Furthermore, the content of C6 esters in the control berries was 40.58% and 13.69% lower than the S1 and S2 berries, respectively, and 27.93% higher than that of the S3 berries at EL 38 (Figure 4f). The content of hexanal and ethyl acetate, the most considerable portion of aldehydes and esters, in the S1 berries were increased by 19.84% and 70.62% at EL 38, respectively, relative to the control (Figure 4g,h).

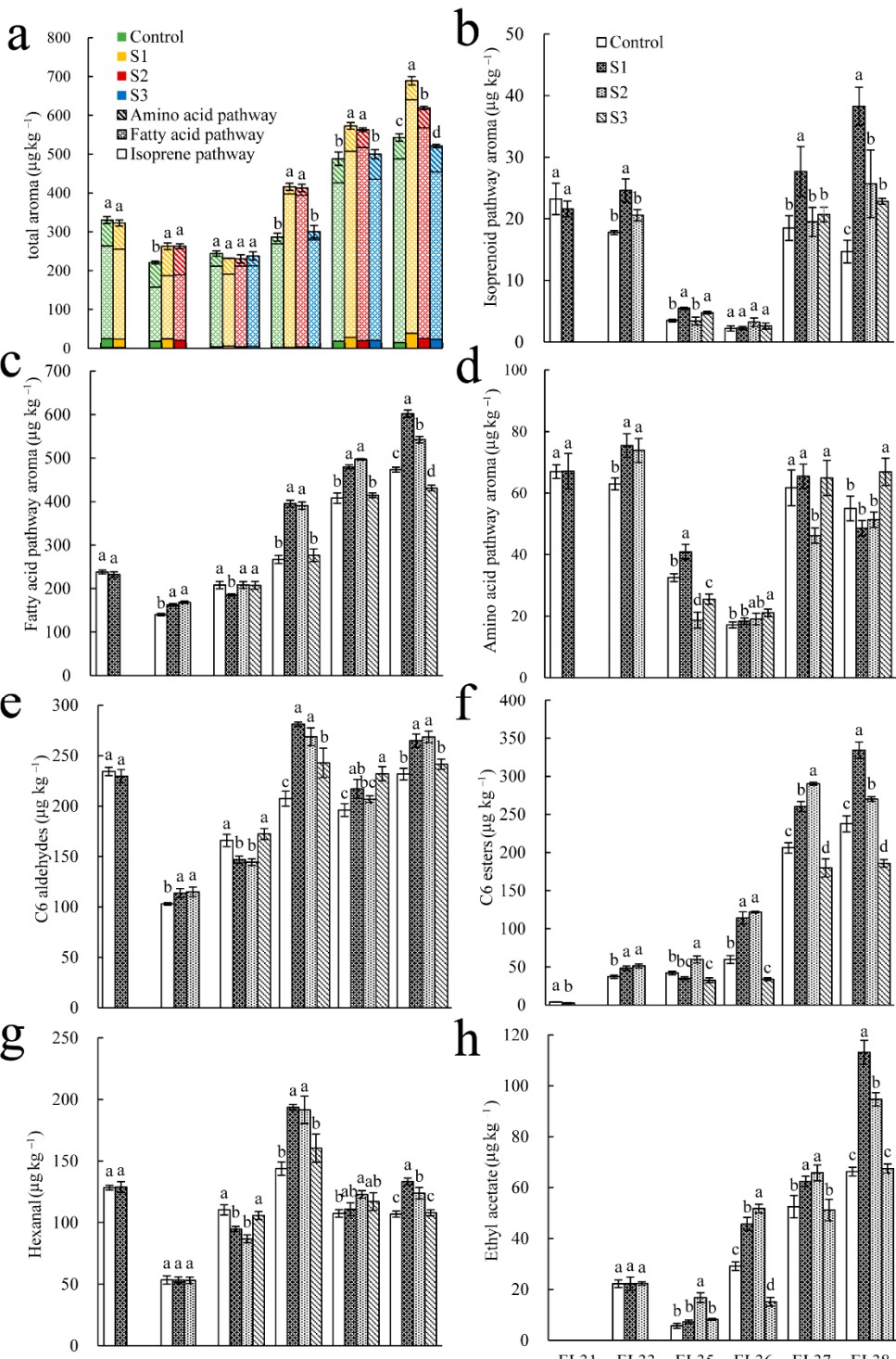

**Figure 4.** Effect of seawater irrigation on aroma in 'Reliance' grape at the growing stage. (**a**) Total aroma, column colors represent different treatments, and column textures represent different aroma pathways (**b**–**d**) changes of aroma in isoprene, fatty acid, and amino acid pathways, (**e**,**f**) changes of total aldehydes and esters in the fatty acid pathway, (**g**,**h**) changes of two main contributors of aldehydes and esters. Values are the means of three replicates, and error bars denote the SD. Different lowercase letters indicate significant differences of samples in one developmental phase, according to Tukey multiple comparison post-hoc test at $p < 0.05$.

### 3.4. Effect of Seawater Irrigation on the Concentrations and Types of Volatiles in 'Reliance' Grape at EL 38

Table 3 shows the effect of 10% seawater irrigation on volatiles originating from different/same precursors and synthetic routes at EL 38. It can be seen that S1 treatment significantly increased most volatiles of the isoprene pathway and C6 esters of the fatty acid pathway. The highest contents of the amino acid pathway volatiles were mostly found in treatment S3. It is worth noting that some volatiles of the isoprene pathway were only detected in the berries irrigated with seawater and were absent in the control berries, such as citronellol, β-myrcene, α-terpineol, and trans-rose oxide (Table 3). The berry volatiles detected in other periods are listed in Tables S3–S7.

**Table 3.** Effect of seawater irrigation on the concentrations and types of volatiles in 'Reliance' grape at EL 38.

| Compounds | RT | RI | *m/z* | Treatment | | | |
|---|---|---|---|---|---|---|---|
| (µg kg⁻¹ FW) | | | | Control | S1 | S2 | S3 |
| Isoprene pathway | | | | 14.68 ± 1.84 c | 38.3 ± 3.06 a | 25.69 ± 5.48 b | 22.84 ± 0.48 bc |
| Terpenoids | | | | 11.44 ± 1.63 c | 33.14 | 21.6 | 18.83 ± 0.25 b |
| Limonene | 12.84 | 1018 | 68, 93 | 5.42 ± 0.97 a | 4.94 ± 0.47 a | 4.79 ± 1.11 a | 5.06 ± 0.09 a |
| 2-Carene | 12.28 | 948 | 41, 121, 93 | 1.55 ± 0.40 a | 1.55 ± 0.53 a | 1.55 ± 0.35 a | - |
| γ-Terpinene | 14.33 | 998 | 93, 136 | 1.33 ± 0.32 a | 1 ± 0.36 a | 1.19 ± 0.30 a | 1.42 ± 0.20 a |
| β-Pinene | 10.36 | 943 | 93, 41, 69 | 1.32 ± 0.36 a | 1.93 ± 0.59 a | 2.79 ± 0.99 a | 3.06 ± 0.63 a |
| Sabinene | 10.28 | 897 | 93, 41, 77 | 1.06 ± 0.24 a | 0.82 ± 0.18 a | 1.82 ± 0.96 a | 1.46 ± 0.44 a |
| α-Pinene | 8.51 | 948 | 93, 77 | 0.76 ± 0.16 a | 1.13 ± 0.05 a | 1.04 ± 0.19 a | 1.3 ± 0.49 a |
| Citronellol | 22.53 | 1179 | 69, 41, 81 | - | 9.57 ± 0.72 a | 3.51 ± 0.98 b | 1.85 ± 0.26 b |
| β-Myrcene | 11.28 | 958 | 93, 41, 69 | - | 2.69 ± 0.62 | - | - |
| α-Terpineol | 20.60 | 1143 | 59, 93, 121 | - | 3.73 ± 0.60 a | 2.59 ± 0.38 a | 2.11 ± 0.87 a |
| 4-Thujanol | 19.93 | 897 | 93 | - | 4.12 ± 1.07 a | 2.31 ± 0.24 b | 2.01 ± 0.28 b |
| Trans-rose oxide | 20.24 | 1114 | 139, 69 | - | 0.97 ± 0.14 | - | - |
| Eucalyptol | 12.97 | 1059 | 43, 81, 71 | - | 0.7 ± 0.09 a | - | 0.56 ± 0.09 a |
| Norisoprenopids | | | | 3.25 ± 0.22 a | 5.15 ± 0.44 a | 4.09 ± 1.35 a | 4.01 ± 0.49 a |
| Methyl heptenone | 11.13 | 938 | 43, 69, 108 | 3.25 ± 0.22 a | 4.29 ± 0.39 a | 4.09 ± 1.35 a | 4.01 ± 0.49 a |
| Geranylacetone | 31.63 | 1420 | 43, 69, 151 | - | 0.86 ± 0.05 | - | - |
| Fatty acid pathway | | | | 473.13 ± 5.60 c | 601.99 ± 8.36 a | 542.33 ± 7.42 b | 430.99 ± 6.59 d |
| C6 alcohols | | | | 3.74 ± 0.85 a | 3.01 ± 0.58 a | 3.57 ± 0.32 a | 3.67 ± 0.40 a |
| 1-Octanol | 15.14 | 1059 | 56, 70, 84 | 0.87 ± 0.39 a | 1.54 ± 0.57 a | 1.37 ± 0.31 a | 1.56 ± 0.20 a |
| 1-Hexanol | 6.19 | 860 | 56, 69 | 2.86 ± 0.46 a | 0.83 ± 0.22 b | 1.31 ± 0.19 b | 0.93 ± 0.22 b |
| 1-Octen-3-ol | 10.79 | 969 | 57, 72 | - | 0.64 ± 0.09 b | 0.89 ± 0.18 ab | 1.18 ± 0.29 a |
| C6 Aldehydes | | | | 231.71 ± 5.79 b | 264.83 ± 6.58 a | 268.53 ± 5.73 a | 241.52 ± 5.07 b |
| Hexanal | 3.92 | 806 | 44, 56 | 106.97 ± 2.46 c | 133.44 ± 2.73 a | 123.88 ± 4.72 b | 108.01 ± 2.46 c |
| (E)-2-Hexenal | 7.47 | 814 | 55, 83, 69 | 95.05 ± 4.49 b | 98.56 ± 4.80 b | 113.04 ± 0.82 a | 95.68 ± 2.33 b |
| Nonanal | 16.63 | 1104 | 57, 70 | 19.38 ± 0.79 b | 21.53 ± 2.37 ab | 22.43 ± 1.96 ab | 23.83 ± 0.80 a |
| Decanal | 21.42 | 1204 | 57, 82, 70 | 7.56 ± 0.53 b | 9.84 ± 0.52 a | 7.38 ± 0.71 b | 8.63 ± 0.76 ab |
| 4-Oxo-2-hexenal | 9.95 | 950 | 83, 55 | 2.26 ± 0.87 a | - | - | 2.41 ± 0.44 a |
| (E)-2-Heptenal | 9.68 | 913 | 41, 83 | 0.49 ± 0.12 b | - | 0.67 ± 0.14 b | 1.31 ± 0.16 a |
| (E)-2-Octenal | 14.41 | 1013 | 70, 55 | - | 1.47 ± 0.18 ab | 1.13 ± 0.19 b | 1.64 ± 0.20 a |
| C6 Esters | | | | 237.69 ± 10.54 c | 334.15 ± 10.87 a | 270.23 ± 2.91 b | 185.8 ± 5.02 d |
| Ethyl acetate | 1.38 | 586 | 43, 70 | 66.3 ± 1.67 c | 113.12 ± 4.71 a | 94.67 ± 2.64 b | 67.48 ± 1.83 c |
| Ethyl butanoate | 4.09 | 785 | 71, 43, 88 | 47.85 ± 2.76 a | 39.07 ± 1.33 b | 41.39 ± 1.16 b | 36.55 ± 3.21 b |
| Ethyl caprylate | 21.12 | 1183 | 88, 57 | 40.65 ± 1.48 b | 46.93 ± 2.33 a | 30.69 ± 0.23 c | 18.75 ± 0.77 d |
| Ethyl 2-butenoate | 5.28 | 793 | 69, 41, 99 | 13.08 ± 0.36 d | 20.49 ± 1.46 c | 34.98 ± 2.01 a | 24.4 ± 1.44 b |
| Ethyl heptanoate | 16.44 | 1083 | 88, 43, 70 | 12.85 ± 1.33 b | 15.94 ± 1.04 a | 10.92 ± 0.07 b | 7.87 ± 0.46 c |
| Ethyl valerate | 7.38 | 884 | 88, 57 | 9.34 ± 0.41 a | 7.49 ± 0.15 b | 6.93 ± 0.14 b | 8.25 ± 1.09 ab |
| Ethyl (E,Z)-2,4-decadienoate | 31.76 | 1397 | 125, 97 | 7.34 ± 1.01 c | 28.02 ± 2.00 a | 15.55 ± 4.75 b | 4.83 ± 0.77 c |
| Ethyl 2-hexenoate | 13.83 | 992 | 97, 55, 73 | 6.5 ± 0.62 b | 12.07 ± 0.13 a | 6.83 ± 0.40 b | 4.86 ± 0.83 c |
| Ethyl caprate | 29.89 | 1381 | 88, 70 | 7.16 ± 1.07 b | 12.3 ± 1.37 a | 4.29 ± 0.16 c | 1.53 ± 0.21 d |
| Ethyl sorbate | 16.22 | 1000 | 67, 95, 41 | 5.91 ± 0.37 a | 3.96 ± 0.72 b | 2.37 ± 0.38 c | 1.61 ± 0.34 c |
| Ethyl (E)-2-octenoate | 23.36 | 1191 | 55, 125, 73 | 4.53 ± 0.93 a | 5.79 ± 0.45 a | 2.55 ± 0.20 b | 1.63 ± 0.56 b |
| Ethyl 4-octenoate | 20.72 | 1191 | 55, 82 | 3.59 ± 0.38 c | 7.6 ± 0.74 a | 5.93 ± 0.65 c | 4.5 ± 0.52 bc |
| Ethyl (E)-4-hexenoate | 12.20 | 992 | 68, 55, 81 | 2.93 ± 0.58 a | 2.65 ± 0.55 a | 2.53 ± 0.29 a | - |
| Ethyl 3-hexenoate | 12.07 | 992 | 69, 41 | 3.15 ± 0.09 ab | 2.75 ± 0.53 ab | 3.4 ± 0.62 a | 2.07 ± 0.38 b |
| Ethyl (E)-4-decenoate | 29.20 | 1389 | 88, 110, 69 | 2.71 ± 0.14 c | 11.21 ± 1.33 a | 5.9 ± 1.28 b | - |
| Ethyl propanoate | 2.23 | 686 | 57, 75 | 1.23 ± 0.54 a | 0.77 ± 0.35 a | 1.29 ± 0.33 a | - |

**Table 3.** *Cont.*

| Compounds | RT | RI | *m/z* | Treatment | | | |
|---|---|---|---|---|---|---|---|
| (µg kg$^{-1}$ FW) | | | | Control | S1 | S2 | S3 |
| Ethyl (E)-2-decenoate | 31.50 | 1389 | 55, 73, 101 | 1.41 ± 0.15 a | 1.17 ± 0.08 a | - | - |
| Ethyl 2-pentenoate | 9.43 | 892 | 83, 55 | 0.6 ± 0.07 | - | - | - |
| Propyl butyrate | 7.35 | 884 | 71, 43, 89 | 0.55 ± 0.11 | - | - | - |
| Butyl butanoate | 12.10 | 992 | 55, 73 | - | 1.29 ± 0.55 a | - | 1.47 ± 0.37 a |
| Ethyl 3-hydroxybutyrate | 8.81 | 947 | 43, 60, 88 | - | 1.53 ± 0.22 | - | - |
| **Amino acid pathway** | | | | 54.97 ± 4.01 b | 48.59 ± 2.48 b | 51.31 ± 2.51 b | 66.86 ± 4.45 a |
| **Benzene derivatives** | | | | 32.41 ± 3.66 b | 30.56 ± 2.34 b | 31.07 ± 1.12 b | 41.78 ± 2.43 a |
| Fluorene | 34.18 | 1494 | 166 | 5.9 ± 0.88 a | 6.38 ± 0.86 a | 5.36 ± 0.88 a | 6.74 ± 0.75 a |
| o-Xylene | 6.03 | 907 | 91, 106 | 5.15 ± 0.86 a | 3.61 ± 0.18 b | 4.96 ± 0.35 a | 4.47 ± 0.29 ab |
| Naphthalene | 20.00 | 1231 | 128 | 4.51 ± 0.53 a | 2.72 ± 0.78 b | 3.88 ± 0.18 ab | 4.98 ± 0.82 a |
| 1,3-Dimethyl benzene | 6.85 | 907 | 91, 106 | 3.12 ± 0.08 a | 2.03 ± 0.57 b | 2.85 ± 0.13 ab | 2.8 ± 0.58 ab |
| Dibenzofuran | 32.90 | 1483 | 168, 139 | 3.06 ± 0.37 b | 4.71 ± 0.80 a | 3.25 ± 0.26 b | 2.8 ± 0.33 b |
| 2-Ethyl toluene | 9.72 | 1006 | 105, 120 | 2.4 ± 0.62 a | - | 2.35 ± 0.21 a | - |
| o-Cymene | 12.67 | 1042 | 119, 134 | 2.29 ± 0.18 a | 2.3 ± 0.18 a | 2.51 ± 0.58 a | 2.43 ± 0.12 a |
| Phenanthrene | 36.95 | 1782 | 178, 76, 152 | 1.74 ± 0.10 a | 1.78 ± 0.39 a | 0.9 ± 0.11 b | 1.86 ± 0.27 a |
| Benzeneacetaldehyde | 13.58 | 1081 | 91, 120 | 1.88 ± 0.48 ab | 3.1 ± 0.17 a | 1.09 ± 0.13 b | 3.48 ± 1.10 a |
| Toluene | 3.13 | 794 | 91, 65 | 0.88 ± 0.14 a | 0.85 ± 0.04 a | 0.91 ± 0.11 a | 0.86 ± 0.08 a |
| Ethylbenzene | 5.74 | 893 | 91, 107 | 0.58 ± 0.18 a | 0.37 ± 0.12 a | 0.69 ± 0.11 a | 0.54 ± 0.08 a |
| Styrene | 6.83 | 883 | 104, 78 | - | 1.73 ± 0.40 | - | - |
| 1-Methylethyl benzene | 13.06 | 804 | 91 | - | - | 1.43 ± 0.49 | - |
| Ethyl benzoate | 19.71 | 1160 | 105, 77 | - | - | - | 9.25 ± 0.68 |
| **Branched volatiles** | | | | 22.56 ± 0.38 ab | 18.02 ± 1.42 c | 20.24 ± 2.37 bc | 25.08 ± 2.02 a |
| 2-Ethyl-1-hexanol | 13.10 | 995 | 57, 70, 83 | 14.21 ± 0.44 b | 17.03 ± 1.28 ab | 16.99 ± 1.32 ab | 19.73 ± 1.67 a |
| 3-Methyl butanal | 1.65 | 643 | 44, 71 | 0.89 ± 0.44 a | 0.99 ± 0.20 a | 0.89 ± 0.56 a | 1.58 ± 0.40 a |
| Isopropyl hexanoate | 13.54 | 1019 | 43, 99, 60 | 6.45 ± 0.32 a | - | 2.36 ± 0.84 b | 2.69 ± 0.36 b |
| 2-Ethyl furan | 2.08 | 742 | 81, 45 | 1 ± 0.20 a | - | - | 1.07 ± 0.21 a |
| **SUM** | | | | 541.89 ± 9.72 c | 687.88 ± 10.79 a | 618.44 ± 4.45 b | 519.11 ± 4.37 d |
| **Varieties** | | | | **Control** | **S1** | **S2** | **S3** |
| **Isoprene pathway** | | | | 7 | 14 | 10 | 10 |
| Terpenoids | | | | 6 | 12 | 9 | 9 |
| Norisoprenopids | | | | 1 | 2 | 1 | 1 |
| **Fatty acid pathway** | | | | 27 | 27 | 26 | 24 |
| C6 Alcohols | | | | 2 | 3 | 3 | 3 |
| C6 Aldehydes | | | | 6 | 5 | 6 | 7 |
| C6 Esters | | | | 19 | 19 | 17 | 14 |
| **Amino acid pathway** | | | | 15 | 13 | 14 | 15 |
| Benzene derivatives | | | | 11 | 11 | 11 | 11 |
| Branched volatiles | | | | 4 | 2 | 3 | 4 |
| **SUM** | | | | 49 | 54 | 50 | 49 |

relative min        relative max

"-" means that the compounds were not detected. Compounds written in bold represent different categories and subclasses of volatiles. Values represent the means ± SD of three replicates. Different lowercase letters within rows indicate significant differences, according to Tukey multiple comparison post-hoc test at $p < 0.05$.

*3.5. Principal Component Analysis of the Effect of Seawater Irrigation on the Volatiles in 'Reliance' Grape during the Berry Development Period*

Principal component analysis (PCA) was carried out to determine the characteristics of the accumulation of volatiles in grapes irrigated with seawater (10%). The first (PC1) and second principal components (PC2) represented 45.47% and 20.342% of the total variance. The volatiles of berries under different growing stages fell into three clusters: volatiles accumulated in the late (EL 37 and EL 38), early (EL 31 and EL 33), and middle periods (EL 35 and EL 36). The three clusters were separated clearly (Figure 5a). The loading plot (Figure 5b) distributed most volatiles, such as α-pinene, limonene, o-cymene, and most esters of the fatty acid pathway (Figure 5c), in the positive half of the *x*-axis (x > 0.5). These volatiles play an essential role in differentiating the EL 37 and EL 38 stages from the other periods. PCA analysis adequately demonstrated the distribution of volatiles from different treatments.

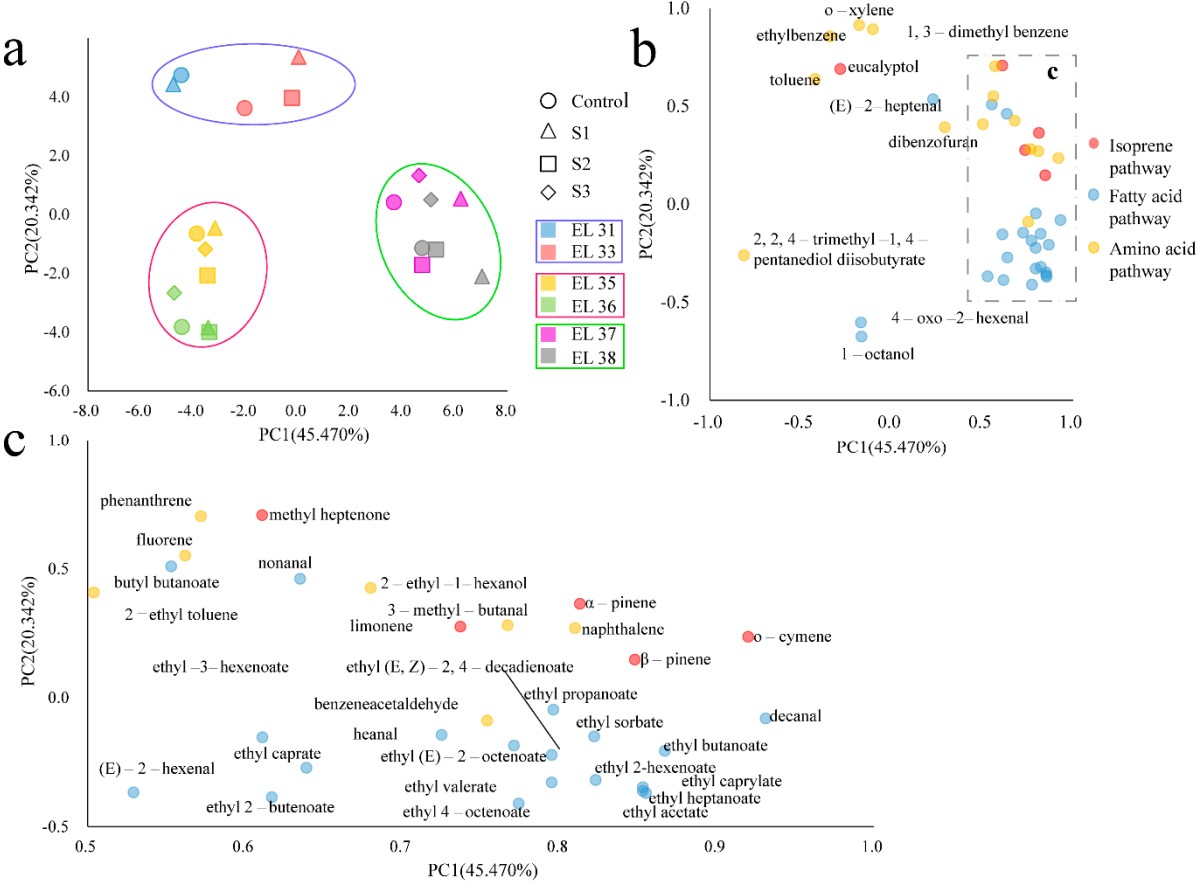

**Figure 5.** Principal component analysis of the effect of seawater irrigation on the volatiles in 'Reliance' grape during the berry development period. (**a**) Score plot of volatiles of berries in different treatment during the grape growing stage. (**b**) Loading plot of the component in a different pathway. (**c**) Specific volatiles of the dotted portion of (**b**).

### 3.6. Relative Expression Levels of Key Genes Responsible for Volatiles in the Fatty Acid Pathway

C6 volatiles, especially C6 esters synthesized from the lipoxygenase-hydroperoxides lyase (LOX-HPL) pathway, produce the primary aroma, contributing to the fruity scent in 'Reliance' grapes. The present study conducted RT-qPCR analysis of *VvLOXA*, *VvHPL*, *VvADH*, and *VvAAT* to describe the relative expression gene profiles of these volatiles. The results revealed similar expression trends during the grape growing period among the four genes (Figure 6). For *VvLOXA*, *VvHPL*, and *VvADH*, the highest expression peaks appeared at either EL 33, EL 35, or EL 36, and then gradually declined until a slight upward trend at EL 38. The expression of the four genes responded most significantly to S1 treatment, which concurs with the trends of C6 volatiles. As the first rate-limiting enzyme in the LOX-HPL pathway, and crucial for fatty acid oxidation and provision of abundant substrates for the synthesis of downstream volatiles, the expression level of *VvLOXA* in the S1 berries was 10.42-fold greater than the control at EL 35. *VvHPL* catalyzes the formation of C6 aldehydes, and the highest expression peak in the S1 berries appeared at EL 33. The *VvAAT* can convert alcohol into C6 esters, and its expression pattern is different from that of the above three genes. Its transcript abundance was relatively low at the early periods, gradually increased from EL 35, and peaked at EL 38. At EL 38, the expression of *VvAAT* in the S1, S2, and S3 berries reached 1.20, 1.99, and 1.68 folds in the control berries, respectively.

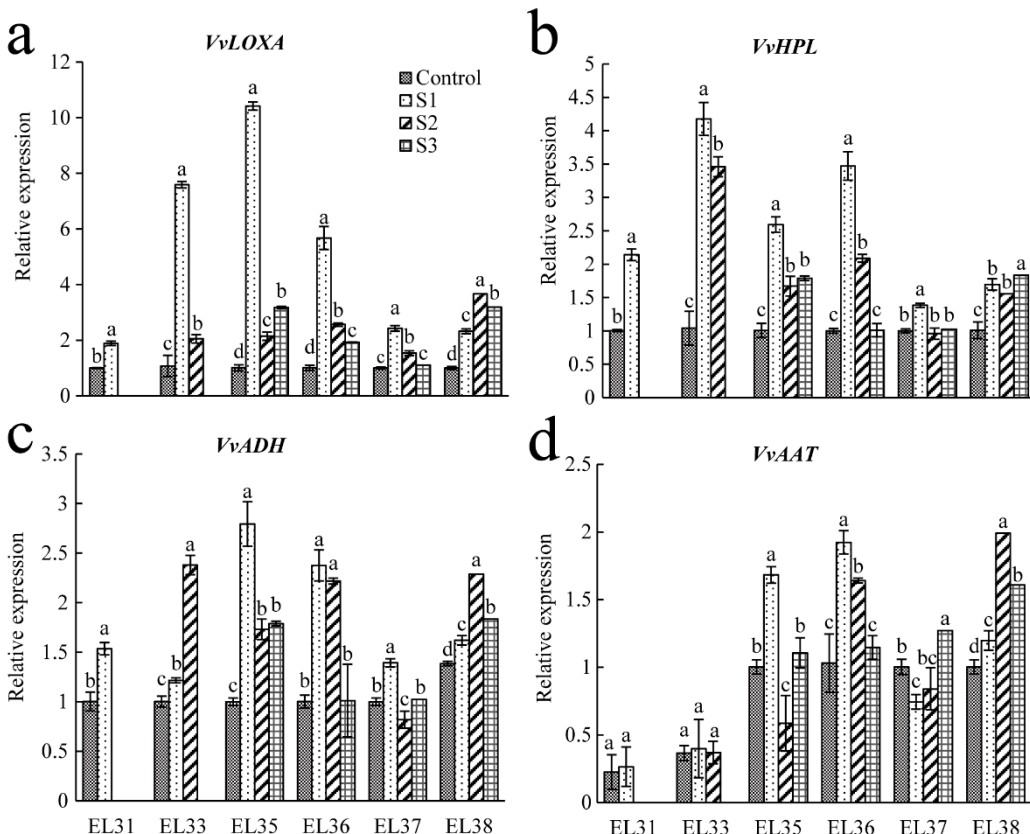

**Figure 6.** Relative expression levels of key genes responsible for volatiles in the fatty acid pathway. (**a**) *VvLoxA*. (**b**) *VvHPL*. (**c**) *VvADH*. (**d**) *VvAAT*. Values represent the means ± SD of three replicates. Different lowercase letters indicate significant differences of samples in one developmental phase, according to Tukey multiple comparison post-hoc test at $p < 0.05$.

### 3.7. Effect of 10% Seawater Treatment on Soil Salinization

The concentration of soluble salt, $Na^+$, $Cl^-$, $SO4^{2-}$, moisture content, and the pH of soil were determined to investigate the effect of 10% seawater irrigation on the soil. Table 4 shows seawater treatment significantly increased the soil pH, especially in the S1 and S2 groups. The soil pH in S1, S2, and S3 treatment groups was 0.21, 0.09, and 0.10 higher than that in the control group. The soluble salt content in the soil after 10% seawater irrigation was also significantly higher than that in the control, but far from the standard measure of soil salinization (1 g kg$^{-1}$). Furthermore, a slight increment in soil $Na^+$, $Cl^-$, $SO4^{2-}$, and moisture content were observed between the control and treatment groups, but the differences were not statistically significant.

**Table 4.** The concentration of soluble salt, $Na^+$, $Cl^-$, $SO4^{2-}$, moisture content, and pH of soil.

| | Control | S1 | S2 | S3 |
|---|---|---|---|---|
| pH | 6.80 ± 0.06 c | 7.01 ± 0.05 a | 6.89 ± 0.02 b | 6.90 ± 0.04 b |
| Soluble salt content (g kg$^{-1}$) | 0.189 ± 0.12 d | 0.704 ± 0.04 a | 0.542 ± 0.08 b | 0.403 ± 0.14 c |
| $Na^+$ content (g kg$^{-1}$) | 0.0158 ± 0.0007 b | 0.0176 ± 0.0008 a | 0.0169 ± 0.0004 ab | 0.0159 ± 0.002 b |
| $Cl^-$ content (g kg$^{-1}$) | 0.0235 ± 0.0005 b | 0.0268 ± 0.0008 a | 0.0261 ± 0.0002 a | 0.0244 ± 0.0001 a |
| $SO4^{2-}$ content (g kg$^{-1}$) | 0.0230 ± 0.0002 b | 0.0270 ± 0.0010 a | 0.0259 ± 0.0005 a | 0.0246 ± 0.0006 a |
| Soil moisture content (%) | 16.13 ± 0.6 a | 17.10 ± 1.3 a | 16.15 ± 0.9 a | 16.55 ± 0.1 a |

Values represent the means ± SD of three replicates. Different lowercase letters indicate significant differences, according to Tukey multiple comparison post-hoc test at $p < 0.05$.

## 4. Discussion

In this experiment, 10% seawater treatment did not alter the average berry weight of the berries, which concurs with the trends of a 4-year seawater irrigation experiment performed by our team [2]. Contrarily, another study showed spraying NaCl (100 mM or 150 mM) on grape leaves increased the fresh weight of the berries. The authors attributed the increment to a rise in relative water content (RWC), mediated by the osmotic regulation of $Na^+$ and $Cl^-$ ions in the berries [7]. The discrepancy between the two results could be because the concentration of NaCl in 10% seawater was so low that osmotic regulation was negligible. Besides, as an essential index for turgor evaluation, the decline in skin hardness and frangibility of the S1 groups at EL 35 indicates that berry softening was promoted, which is a sign of fruit-ripening. Moreover, berries in the S1 group had greater firmness and hardness when compared with the control group berries in the last period, EL 38, suggesting the fruits from the S1 group could have better storage and transport attributes.

Sugar and organic acids are basic indexes of grape-ripening. They also determine the organoleptic quality and flavor of table grapes [7]. It is postulated that salt-treated cherry tomatoes have higher titratable acid concentrations and reduced sugars than the controls [6]. It is also postulated that water with high electrical conductivity (EC) can enhance the TA and TSS content of tomato [23]. Herein, the concentration of TSS, glucose, and fructose in treated grape berries was increased at the EL 35 stage. However, malic and tartaric acid content had a decreasing trend compared to the control. The trends of titratable acidity and TSS levels were consistent with those of a 4-year seawater treatment for grapevines [2]. Based on the disparities of these results, the influence of salt treatment on grape quality could have been affected by the various salt concentrations used. The significant changes in glucose and fructose levels due to seawater treatment indicate that environmental changes easily affect sugar concentrations and maintain osmotic pressure.

As an essential determinant of grape quality, volatiles exist as free or bound glycosides. Among them, the free forms could be more vital ingredients that determine the flavor, although odorless bound glycosylated forms could be hydrolyzed to odor-active free forms during fermentation [24]. C6 esters and terpenes are the main contributors to the abundant fruity and negligible flower scent in 'Reliance' grape. The aroma in different pathways first reduced and then increased (Figure 4). The turning point was during the fruit expansion period because the concentration of volatiles in the fruit may be diluted, which concurs with previous results [25]. Furthermore, since C6 volatiles (also called GLVs) were more sensitive to environmental changes [26], the change in the concentration of its volatiles occurred earlier than the volatiles of the other two pathways. Previous studies in strawberry [27] and Nero d'Avola wine [28] found that moderate and low salinity levels increased ester concentrations consistent with the trend observed in this experiment.

In this experiment, aroma content related to the isoprene pathway at EL 31 reached 23.22 μg kg$^{-1}$ (Table S4). It was mainly eucalyptol (63.43%), with a "eucalyptol" and "medicinal" flavor, that was found in 'Reliance', 'Riesling', and Australian 'Cabernet sauvignon' grapes but not in the Shine Muscat grape [29]. In contrast, Wu reported that isoprene pathway volatiles, especially terpenes, are the main contributors of the flower scent, and their concentrations were substantially low before veraison [25]. The divergence between our findings and that of Wu et al. (2020) can be attributed to differences in the aroma features of various grape cultivars. Compared with the control berries, more terpenes, such as citronellol, β-myrcene, α-terpineol, and trans-rose oxide, were detected in the S1 berries (Table 3); these low threshold volatiles can provide floral and rose scents to the 'Reliance' grape. However, 10% seawater treatment did not alter the aroma of the amino acid pathway volatiles, suggesting the grape cultivars characterized by the aroma from the amino pathway may not benefit from 10% seawater irrigation.

The developmental period-specific evolution of grape volatiles during ripening has been reported widely. In 'Cabernet Sauvignon' grapes, C6 volatiles were mainly accumulated in the form of C6 aldehydes, C6 alcohols, and C6 esters during the veraison period, post-veraison period, and maturity period [30]. In the Muscat grape 'Shine Muscat' (Vitis

labrusca × V. vinifera), C6 aldehydes were synthesized in large quantities at the veraison stage, C6 alcohols at the post-veraison stage, and C6 esters and terpenes during early maturity [31]. Though irrigation with 10% seawater significantly affects some key volatiles (Table 3), the accumulation of volatiles mainly follows the berry development patterns.

The expression levels of the four structural genes in the LOX-HPL pathway were closely related to the C6 volatile generation. *VvLOXA* was the gene responsible for changes in 'Reliance' grape volatiles under 10% seawater treatment. Previous studies postulate that plants release C6 volatiles in response to biotic stresses [17]. The remarkable upregulation is evident in many LOX genes under salt stress in tomato (*Solanum lycopersicum* L.) [32]. Herein, *VvLOXA* showed two consecutive transcript abundance peaks at EL 33 and EL 35 as the first rate-limiting gene in the LOX-HPL pathway (Figure 6a). This finding was attributed to *VvLOXA* activation, which consequently oxidized fatty acids and provided substrates for the synthesis of volatiles in response to 10% seawater irrigation, especially in the S1 treatment. The highest peak of *VvHPL* expression and content of C6 aldehydes appeared at EL 33 and EL 36, respectively, revealing a delay between changes in the genes and volatiles under seawater treatment. In addition, water deficits also increased the transcription level of *VvLOX* and *VvHPL* in 'Cabernet Sauvignon' grapes, thus, enhancing the metabolism of C6 volatiles [33,34]. Notably, the expression levels of *VvADH* and *VvAAT* in S2 and S3 were higher than those of the S1 berries at EL 38, with the concentrations of C6 esters showing an opposite trend. This result was attributed to the high transcript levels of *VvLOXA* and *VvHPL* in the S1 berries, which resulted in more abundant C6 aldehydes for the synthesis of esters. Based on the relative expression gene profiles of these volatiles, we concluded that *VvLOXA* is the key gene that alters GLVs composition in grapes. This result was consistent with that of a previous study in which an increasing pattern of the *VvLOXA* transcript at the pre-veraison stage was observed in four grape varieties for two vintages [14].

Seawater irrigation has a long history; in Israel in 1966, Hogo Boyko creatively mixed freshwater and seawater on the sand to irrigate crops [35]. Irrigating crops with undiluted seawater could cause soil salinization, nutrient loss, and damage to the soil's physical structure, which are not conducive to plant growth, but in this experiment, the soluble soil salt content under the S1 and S2 treatments did not reach 1 g kg$^{-1}$ and was, thus, far from causing soil salinization. This low salinity could be due to the heavy seasonal rainfalls and moderate salinity of 10% seawater. On the other hand, large-scale soil acidification due to fertilizer overdose is a current challenge in China. The application of seawater irrigation could, therefore, alleviate this soil acidification gradually. In summary, the 10% seawater irrigation of grapevines during the whole growing period is a sustainable and feasible strategy and does not cause soil salinization.

## 5. Conclusions

Seawater irrigation significantly increased the sugar content but decreased the organic acid level in grape berries compared to the controls. S1 treatment significantly increased the concentrations of C6 esters and the terpene levels (citronellol, β-myrcene, α-terpineol, and trans-rose oxide) in berries. Berries irrigated with 10% seawater had better fruit quality during the pea-size stage (S1) than S2, S3, and the control groups. RT-qPCR analysis revealed that *VvLOXA* is the key gene involved in volatile biosynthesis when grapes are treated with 10% seawater. Combined with the 4-year 10% seawater irrigation experiment results obtained by our team, irrigation with 10% seawater since the pea-size period is a feasible measure in viticulture without causing soil salinization. Further, it is worthwhile to explore the effects of seawater irrigation on volatiles, phenolic acids, and anthocyanin compounds in areas with annual rainfall greater than 638 mm.

**Supplementary Materials:** The following supporting information can be downloaded at: https://www.mdpi.com/article/10.3390/horticulturae8030248/s1, Table S1: Meteorological data for the vineyard in 2018; Table S2: Calibration curves for the quantification of volatile compounds; Table S3: Primer sequence for RT-qPCR; Table S4: Effect of seawater treatment on the concentrations and

varieties of volatiles in 'Reliance' grape at EL 31; Table S5: Effect of seawater treatment on the concentrations and varieties of volatiles in 'Reliance' grape at EL 33; Table S6: Effect of seawater treatment on the concentrations and varieties of volatiles in 'Reliance' grape at EL 35; Table S7: Effect of seawater treatment on the concentrations and varieties of volatiles in 'Reliance' grape at EL 36; Table S8: Effect of seawater treatment on the concentrations and varieties of volatiles in 'Reliance' grape at EL 37.

**Author Contributions:** Conceptualization, H.Z.; Data curation, M.L.; Formal analysis, M.Y.; Funding acquisition, Y.D.; Project administration, M.L. and H.Z.; Resources, M.T.; Software, M.Y.; Visualization, Y.Y. and Y.D.; Writing—original draft, M.L.; Writing—review and editing, Z.G. All authors have read and agreed to the published version of the manuscript.

**Funding:** This research was supported by China Agriculture Research System of Ministry of Finance (MOF) and Ministry of Agriculture and Rural Affairs (MARA) of the People's Republic of China (Funding Number: CARS-29).

**Institutional Review Board Statement:** Not applicable.

**Informed Consent Statement:** Not applicable.

**Data Availability Statement:** Not applicable.

**Conflicts of Interest:** The authors declare no conflict of interest.

## Abbreviations

GLVs: green leaf volatiles; MPs, methoxypyrazines; TSS, total soluble solids; TA, titratable acid; ABA, abscisic acid; EC, electrical conductivity; RWC, relative water content; FW, fresh weight; RT-qPCR, Real-time quantitative PCR assays.

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
