# Peer review of "Effect of Seawater Irrigation on the Sugars, Organic Acids, and Volatiles in ‘Reliance’ Grape"

_horticulturae, doi:10.3390/horticulturae8030248_

Round 1
Reviewer 1 Report
In this manuscript the authors investigated the effects of seawater irrigation on the sugars, organic acids and volatiles in ‘Reliance’ grape production. The manuscript is well prepared and it may be accepted for publication in Horticulturae journal, after the manuscript will be revised regarding the following comments.
Line 15: Could you be more precise for ‘berry expansion period’?
Line 16: Mention also the control treatment.
Lines 68-69: Instead of ‘since different stages’ write ‘starting at different grapevine phenological stages’.
Lines 76-78: Briefly describe the type of climate (also concerning the temperatures) because all the presented grapevine phenological stages appeared very early and it may be supposed that the climate is very warm.
Lines 86-90: Be more precise and detailed in describing the irrigation treatments. Were the vines irrigated during these phenological stages (as stated in line 86, possibly wrong), or starting at these phenological stages? Were the vines irrigated every seven days as stated in the Abstract? Were the vines irrigated up to harvest, or? Be more precise for the ‘expand period’ (S2). Specify the quantity of water used for irrigation. Based on which data was this quantity defined? For the control treatment specify the initial stage of irrigation.
Line 97: delete ‘concentration’. Do you have the physical and chemical properties also for the fresh water?
Line 99: specify for what were used these samples.
Figure 3.a): It is strange that total soluble solids were around 10% at stage EL33, as they are usually below 5 at this stage. Are you sure that you properly determined the phenological stages?
Figure 4.a): Why it is used the color here?
Line 367: replace ‘did not cause’ with ‘on’.
Lines 462-467: Move this text to the Introduction section, or rewrite the text appropriately for the discussion section.
Reviewer 2 Report
The paper is very well prepared and documented. The results are clearly described. The conclusion are stated very well and comes from results and discussion. I am under impress of the study and type of analysis, however I have several issues:
- It is very good that you measure the quality of seawater used in this study and ions accumulation in the soil. Did you monitored also the EC value of the soil. Such results could be very informative in case of eventually stress effect. This is just a suggestion.
- As not only the water and salt can influence the level of sugar and other molecules present in fruits it should be good to enhance the paper in record from the nearest meteorological station. You have a control plants, which were irrigated with fresh water, but such results could be useful from agrotechnical point of view. You can add the report in supplementary files. This is just a suggestion.
- If you presented all the results for developmental stages at 1 graph and mark with the letters a, b, c… all the samples it suggested that there is no differences between particular probes with the same letter. As in Figure 1 f.eg. between developmental phases, as EL31 and EL38. You should used other test to compare samples also between phases or properly describe the figures. This should be corrected.
- Why Fig 4a is in color and the rest no? This is just a question.
Reviewer 3 Report
Please find my comments into the attached doc. Congratulations for your research paper !

Author Response
Line 11. I can't find the logic of this sentence that includes two distinct ideas. Please reformulate.
RE: Thanks for your suggestion, I revised it to “Ongoing climate change in recent decades exacerbated the decline in agricultural water use, and seawater irrigation could feasibly alleviate the shortage of water resources, which restricts viticulture in some countries.” Line 11
Line 14. it is necessary to fill with "diluted" sea water (10% concentration)
RE: Done. Line15
Round 2
Reviewer 1 Report
The manuscript has been improved according to reviewers suggestions and I suggest to publish it in Horticulturae journal.